# Surveillance of Antimicrobial Resistance in the ECOWAS Region: Setting the Scene for Critical Interventions Needed

**DOI:** 10.3390/antibiotics13070627

**Published:** 2024-07-05

**Authors:** Ahmed Taha Aboushady, Olivier Manigart, Abdourahmane Sow, Walter Fuller, Abdoul-Salam Ouedraogo, Chinelo Ebruke, François-Xavier Babin, Laetitia Gahimbare, Issiaka Sombié, John Stelling

**Affiliations:** 1Brigham and Women’s Hospital, Harvard Medical School, Boston, MA 02115, USA; jstelling@whonet.org; 2West African Health Organization, Bobo-Dioulasso 01 BP 153, Burkina Fasoabdourahmane.sow@pasteur.sn (A.S.); chinelo.ebruke@gfa-group.de (C.E.); isombie@wahooas.org (I.S.); 3GFA Consulting Group, 22359 Hamburg, Germany; 4Ecole de Santé Publique, Université Libre de Bruxelles, 1050 Brussels, Belgium; 5Institut Pasteur de Dakar, Dakar 220, Senegal; 6Department de Santé Public, Faculté de Médecine, de Pharmacie et D’Odontostomatologie, Université Cheikh Anta Diop, Dakar 5005, Senegal; 7World Health Organization Regional Office for Africa, Brazzaville P.O. Box 06, Congo; wfuller@who.int (W.F.); gahimbarel@who.int (L.G.); 8Centre Muraz, Institut National de Santé Publique, Bobo-Dioulasso 01 BP 390, Burkina Faso; abdousal2000@yahoo.fr; 9Fondation Mérieux, 69002 Lyon, France; fx.babin@fondation-merieux.org

**Keywords:** antimicrobial resistance, surveillance, West Africa, ECOWAS, infectious diseases

## Abstract

Antimicrobial resistance poses a significant challenge to public health globally, leading to increased morbidity and mortality. AMR surveillance involves the systematic collection, analysis, and interpretation of data on the occurrence and distribution of AMR in humans, animals, and the environment for action. The West African Health Organization, part of the Economic Community of West African States (ECOWAS), is committed to addressing AMR in the region. This paper examines the status of AMR surveillance in ECOWAS countries using available WHO data from the TrACSS survey and GLASS enrollments. The analysis reveals that while progress has been made, significant challenges remain. Twelve of the fifteen ECOWAS countries are enrolled in GLASS, and ten have developed national action plans (NAPs) for AMR. However, there is a need to ensure all countries fully implement their NAPs, continue reporting to GLASS, and use the data for evidence-based actions and decision making. Surveillance systems for AMR and antimicrobial consumption/use vary across countries with some demonstrating limited capacity. All countries, except Cabo Verde, reported having a reference laboratory for AMR testing. Strengthening laboratory capabilities, data management and use, and multisectoral coordination are crucial for effective AMR surveillance and response. Based on the findings and the regional context, it is essential to prioritize capacity building, data utilization, and the adoption of standardized guidelines for AMR surveillance. Collaboration among ECOWAS countries, the WAHO, and international partners is essential to address AMR comprehensively. Ensuring a consistent supply of essential antimicrobial medications and reagents is vital.

## 1. Introduction

Antimicrobial resistance (AMR) poses a significant challenge to public health globally, leading to increased morbidity, mortality, and healthcare costs [1,2]. AMR hampers the effectiveness of treatments for severe infections, contributing to millions of deaths each year [3]. According to a predictive model, bacterial AMR was associated with 4.95 million deaths in 2019, and the most significant burden occurred in the sub-Saharan Africa region, where 1.07 million people died because of bacterial antimicrobial resistance [4]. The burden of AMR is not evenly distributed, and resource-limited countries face a higher impact [5,6,7]. In Africa, the risks associated with AMR are amplified by the high prevalence of infectious diseases and fragile healthcare systems [8,9]. Furthermore, this is exacerbated by the misuse and overuse of antibiotics, including non-prescription use, which is widespread in some parts of Africa and exceeds 90% of all antimicrobial use in some parts of Africa [10]. While considerable progress has been made in understanding the drivers of AMR and implementing antibiotic stewardship programs, low- and middle-income countries have often been left behind from such advancements.

In May 2015, The Global Action Plan (GAP) on Antimicrobial Resistance (AMR) [11] was adopted by all WHO Member States through a resolution in the World Health Assembly and endorsed by the Food and Agriculture Organization of the United Nations (FAO) Governing Conference and the World Assembly of World Organization for Animal Health (WOAH, founded as OIE) delegates. Strengthening the evidence base through enhanced global surveillance and research is one of the five objectives of the GAP. With the endorsement of the GAP, countries agreed to develop a national action plan (NAP) on AMR and to implement relevant policies and plans to prevent, control, and monitor AMR. Countries also noted their commitment to completing the annual Tracking AMR Country Self-Assessment Survey (TrACSS) to monitor progress in NAP implementation. The survey name was changed in 2022 from ‘Tripartite’ to ‘Tracking’ AMR Country Self-Assessment Survey to reflect the inclusion of the UN Environment Program (UNEP) within the Quadripartite (FAO, UNEP, WHO, WOAH). Since 2016, the number of countries participating in the survey has increased with more than 80% of AFRO countries participating in TrCASS 2022.

AMR surveillance plays a crucial role in addressing the growing AMR problem. AMR surveillance involves the systematic collection, analysis, and interpretation of data on the occurrence and distribution of AMR in humans, animals, and the environment [12]. It provides valuable insights into AMR trends, patterns, and factors, enabling evidence-based decision making for implementing appropriate measures [13]. However, many African countries face challenges in establishing robust surveillance systems and collecting comprehensive prevalence data [14]. According to a systematic review of AMR by the WHO Regional Office for Africa in 2021, there is a significant lack of data on the prevalence of AMR in the WHO African Region, particularly in West Africa. This is primarily due to the region’s limited laboratory capacity and surveillance networks [15]. Nevertheless, two cohort studies from Ghana have shown that AMR is associated with an additional mean healthcare providers’ annual cost of 650,000 USD and a patient cost of 1.4 million USD, calling for the prioritization of the AMR prevention surveillance, and mitigation in hospitals and the community [16,17]. According to the World Bank’s modeled estimates, AMR health costs could increase to $330 billion under a low burden and $1.2 trillion under a low burden [18].

Upon adoption of the GAP, Member States requested WHO to help with the establishment of a Global Antimicrobial Resistance and Use Surveillance System (GLASS) that supports a standardized approach to the collection, analysis, and sharing of AMR, antimicrobial consumption (AMC) and antimicrobial use (AMU) data, promotes the One Health model for AMR surveillance, and generates data to support patient care, inform policies, strategies, and AMR burden estimates [19]. GLASS also contributes to informing the progress in achieving Sustainable Development Goals (SDGs) [20], especially SDG3 [21] with two indicators: the proportion of bloodstream infections (BSIs) among patients seeking care due to methicillin-resistant *Staphylococcus aureus* (MRSA) and *Escherichia coli* resistant to third-generation cephalosporins. By participating in GLASS, countries commit to build or strengthen their national AMR surveillance systems to generate quality AMR surveillance data to meet local needs and GLASS requirements and share these data globally. In 2020, GLASS incorporated the antimicrobial consumption (GLASS-AMC) surveillance module, which provides a basis for countries to understand their respective consumption level and use that to put in place appropriate policies, regulations, and interventions for the responsible use of antimicrobials. The 5th GLASS report was published in December 2022 [22]. As of September 2023, 130 countries, territories, and areas are registered worldwide, including 38 out of 47 countries of the WHO African Region [23].

Closing the gaps in AMR surveillance and strengthening capacity in Africa is crucial for early detection and response to AMR outbreaks [24,25]. By establishing effective surveillance systems and collecting comprehensive data, policymakers and healthcare professionals can make informed decisions to combat AMR, improve patient outcomes, and safeguard public health [26]. Effective AMR surveillance relies on robust laboratory capacity to accurately identify and characterize resistant pathogens, standardized data collection and reporting methodologies, and interoperable information systems enabling health authorities and stakeholders to gain data sharing and analysis [13]. The data collected through surveillance systems provide a comprehensive understanding of the current AMR situation burden of disease, guide the development of antimicrobial stewardship programs, aid in the early detection and response to outbreaks or emerging resistance threats, and guide the development of appropriate antimicrobial stewardship programs that are relevant to the local setting [27,28,29].

The Economic Community of West African States (ECOWAS) is a regional organization comprising 15 member countries in West Africa [30]. The ECOWAS aims to promote economic integration and stability within the region. One of the seven key ECOWAS institutions is the West African Health Organization (WAHO), which focuses on improving the health and well-being of people in West Africa [31]. One significant area of concern for the WAHO is antimicrobial resistance (AMR). The WAHO has taken proactive measures to combat AMR and promote antimicrobial stewardship in the region. The WAHO’s initiatives in addressing AMR include strengthening surveillance systems, enhancing laboratory capacity, and promoting the rational use of antimicrobials in healthcare settings. The WAHO works closely with member countries, regional partners, and international organizations to develop and implement effective policies and strategies to tackle AMR [32,33]. The WAHO adopts a One Health approach, recognizing the interconnectedness of human, animal, agricultural, and environmental health in addressing AMR [31].

The ECOWAS, in its vision for the region known as “ECOWAS Vision 2050”, acknowledges the importance of a resilient and sustainable health system [34]. This vision aligns with the WAHO’s commitment to addressing AMR and other health challenges in West Africa. By fostering collaboration and knowledge sharing, ECOWAS and WAHO aim to strengthen healthcare systems, improve access to quality healthcare services, and mitigate the impact of AMR on the region’s population [32,34].

The COVID-19 pandemic has impacted AMR in multiple ways, including reversing some of the progress made in the fight against AMR worldwide. [35,36]. Antibiotics, especially broad-spectrum ones, were overprescribed due to challenges distinguishing bacterial infections from COVID-19, which was exacerbated by the use of medical devices and shortages of personal protective equipment [37]. The pandemic also underscored the importance of improving AMR management with increased awareness of hygiene practices and infection prevention and control measures and the need to strengthen laboratory capacities and surveillance systems [37,38]. The pandemic also highlighted the necessity to optimize antibiotic use, including appropriate prescribing, rapid diagnostics, and adherence to treatment guidelines [37]. Additionally, vaccinations can reduce the need for antibiotics, and continued investment in AMR research, surveillance, and stewardship programs is necessary to address this growing threat [37].

Through this publication, we aim to utilize publicly available information to describe the status of AMR surveillance in the West African region and identify opportunities for improvement. We are bringing together different global, regional, and in-country expertise to analyze the situation and to put forward surveillance strengthening and investment recommendations.

## 2. Results

### 2.1. Country Background Information

Country responses to TrACSS and background information for West African countries are shown in Table 1.

### 2.2. National Action Plan (NAP) Status

A review of the national action plans (NAPs) showed that the Guinea-Bissau NAP is still being developed. Three countries, namely Burkina Faso, Niger, and the Gambia, have finalized NAPs that are yet to be endorsed. In contrast, all other countries have finalized and endorsed NAPs, including a monitoring and evaluation plan. The complete information on the countries’ statuses can be found in Figure 1a.

### 2.3. GLASS Enrollment Status

The data extracted from the different sources were used to visualize the status in West Africa. Regarding GLASS 2022 status, except for three countries, Senegal, Guinea, and Guinea-Bissau, all countries in West Africa have been enrolled. Apart from the three countries mentioned, all countries in the region are enrolled in at least one GLASS-AMR module with five countries enrolled in GLASS-AMR and AMC modules. The complete information on the countries’ enrollment statuses can be found in Figure 1b.

### 2.4. Surveillance Systems

Overall, the region’s country capacity for AMR surveillance appears fairly developed, with 14 of the 15 countries reporting some AMR surveillance capacity. Of the countries with some level of AMR surveillance capacity, six countries have limited AMR surveillance capacities, three have developed capacities, and five have demonstrated capacities.

In contrast to the findings on country capacity for AMR surveillance, country capacity for AMC/U surveillance in the region is much less developed. Only two countries, Mali and Ghana, have demonstrated capacity for AMC/U surveillance, while six have limited capacity, and five have no AMC/U surveillance capacity. There were no responses from two countries. The full breakdown by country and their status can be found in Figure 2.

### 2.5. Surveillance System Capacities

Within the human sector, the AMR surveillance capacities have improved despite the pandemic hindering many health services. Of the ten countries that responded in 2018 and 2022, all ten could generate AMR data in 2022 compared to only nine in 2018. Sierra Leone was the one country declaring improvement between 2018 and 2022. Also, three more countries, Burkina Faso, Cote d’Ivoire, and Ghana, succeeded between 2018 and 2022 in establishing a functioning national AMR surveillance system with external quality assessment and a national coordinating center producing reports, reaching five countries in 2022. Responses for the whole regional situation can be found in Figure 3; responses for individual countries can be found in Appendix A.

Regarding the country’s capacity for detecting the WHO priority pathogens in the TrACSS survey, all countries in the region except one (Cabo Verde) have a national reference laboratory. Reference laboratories in eight countries can perform antimicrobial susceptibility tests for 11 priority pathogens (*Acinetobacter baumannii*, *Pseudomonas aeruginosa*, *Escherichia coli*, *Klebsiella*, *Proteus*, *Enterococcus faecium*, *Staphylococcus aureus*, *Campylobacter* spp., *Salmonella* spp., *Neisseria gonorrhoeae*, *Streptococcus pneumoniae*, *Haemophilus influenzae*, and *Shigella* spp.). In addition, reference laboratories in four other countries can test 10 priority pathogens, with Campylobacter spp. being the most common challenging organism. There was incomplete or no response from the two countries. Full results by country can be found in Appendix A.

Furthermore, as in Figure 4, only two countries have a mechanism to report stockouts at the national level, and only one has a mechanism to report stockouts at the local level. In comparison, all other nine countries have no central mechanism for reporting stockouts. The number of days of stockout disrupting the services of the National Reference Bacteriology/Antimicrobial Susceptibility Testing (AST) Laboratory over the last three months being reported was variable. Two countries reported 90 days (stockout for the entire period in question), two reported 60 days (stockout for two thirds of the period in question), and one country reported 45 days (stockout for the half period in question).

### 2.6. Integration and One Health

No country responded as having adequate technical capacity, resources, or established systems to collect data across all relevant sectors of One Health—human, animal, environmental, and food. Furthermore, only three countries are analyzing data collected and using them for decision making, advocacy, policy changes, and the allocation of adequate resources through the AMR multisector coordination mechanism. AMR’s integration into other action plans and strategies has varying extents related to the different One Health sectors. Regarding the integration into the national One Health strategies, only nine out of 14 countries have a One Health strategy that includes AMR. The complete information can be found in Appendix A.

### 2.7. Data Use

Six of the fourteen countries use the data they collect for AMR and AMC/U to inform operational decision making and amend policies.

## 3. Discussion

The GLASS and NAP status analysis in West Africa reveals critical insights into the region’s current AMR surveillance and action plans. For the twelve among the fifteen ECOWAS countries currently enrolled in GLASS, these countries are highly encouraged to continue reporting to GLASS annually and enroll in several GLASS modules [22,27,40,41]. Additionally, all committed parties should support the three remaining countries enrolling in GLASS modules to strengthen regional surveillance efforts. The data should not stop here, but countries should use available data to inform policy and decision making.

Regarding NAPs, most countries have developed their plans and should be implementing and possibly renewing them soon. However, Guinea-Bissau is still developing its NAP, and three other countries have finalized theirs but pending endorsement. This confirms the necessity for more support and collaboration within and for the region to ensure the timely implementation of effective AMR action plans [42]. The NAPs should follow the WHO NAP implementation guidelines, including mobilizing a One Health multi-stakeholder group and using the WHO costing and budgeting tool [43,44,45]. Nevertheless, it is important to ensure that these costed plans, including an investment case, are used internally and externally for resource mobilization.

The analysis of surveillance systems reveals a mixed picture. While some countries have demonstrated capacity in AMR and AMC/U surveillance, many still have limited or no capacity. This is alarming and requires prioritizing strengthening surveillance systems at institutional and/or national levels, particularly for AMC/U, which is less developed than AMR surveillance [46,47,48,49]. Efforts should be focused on providing technical support and resources to enhance surveillance capacities in these countries [12].

Integrating AMR efforts across human, animal, food, and environmental sectors is critical to addressing the complex challenge of AMR [50,51]. However, the findings indicate that no country in West Africa reported having adequate technical capacity, resources, and established systems to collect data across all relevant sectors. The available data on this study do not exactly explain this current scenario. However, the results may indicate a disparity in capacity for AMR surveillance and, consequently, for generating AMR data across the different sectors. This could point to the need to strengthen the AMR surveillance capacity in sectors other than the human sector to ensure the optimal cooperation and effective integration of AMR data across sectors. This approach will foster effective partnership and collaboration across sectors and facilitate data collection, sharing, and analysis for evidence-based decision making. Only a few countries are currently analyzing the data collected and using them through AMR multisector coordination mechanisms. This suggests the need for strengthening such mechanisms and advocating for policy changes and resource allocation based on the data [52]. Integrating AMR into action plans and cognate strategies of the different One Health sectors is also challenging. The varied responses of countries in this regard indicate a lack of consistent integration of AMR considerations into broader health and development agendas. This highlights the importance of advocating for the greater recognition of AMR as a cross-cutting issue and the need for more robust integration within existing policies and strategies.

Looking into the laboratory capacities, only Cabo Verde has no national reference laboratory. However, they have two bacteriology laboratories performing microbial cultures, species identification, and antimicrobial susceptibility testing. This is particularly important to implement quality assurance and serve as the main facility to conduct nationwide antimicrobial susceptibility testing (AST) on first and second-line drugs, especially given their increasing rate of antimicrobial drug resistance [53]. Furthermore, several countries reported that they could isolate all priority organisms except Campylobacter spp. [54]. Campylobacter is a fastidious organism. Therefore, it is challenging to cultivate due to its exceptional adaptation to the host it infects, requiring rapid processing and specialized incubation conditions for growth [54].

Moreover, the findings on the use of data collected indicate room for improvement in leveraging the collected data for decision making and policy amendment. Only a few countries utilize the data for operational decision making and policy amendments. This emphasizes the need to enhance data utilization capacities and promote a culture of evidence-based decision making to drive effective AMR interventions.

## 4. Materials and Methods

This review encompasses the status of AMR surveillance in ECOWAS countries. Data were obtained from the Global Database for Tracking Antimicrobial Resistance Country Self-Assessment Survey (TrACSS) [55], from GLASS reports, and from AFRO records on NAP development, implementation, and monitoring [23]. It provides global benchmarking for country self-assessment for all sectors and general global messages for country action on AMR from the quadripartite [56].

Data for 2022 were extracted from the TrACSS database, including information on AMR surveillance activities conducted by participating countries. Data from participating ECOWAS countries were included in the analysis. The latest responses from 2018 represented the status before the COVID-19 pandemic.

The TrACSS survey requests that each country submits an official response, validated by all relevant sectors, summarizing national progress. It recommends engaging a multisectoral group collaboratively to ensure comprehensive and accurate responses. The ideal forum for this coordination is a national AMR coordinating committee or a multisectoral working group on AMR. In cases where such structures are absent, officials from relevant ministries should collaborate to determine the response process. Most questions in the survey use a five-point rating scale (A to E) to assess national capacity and progress with some using a four-point scale (A to D) when less variation is expected. These ratings align with capacity levels in the International Health Regulations (IHR) questionnaire and reflect progress and functionality. The response reflects the closest match to the country’s situation, and higher ratings should encompass the progress levels of lower ratings.

Furthermore, data from the Global Antimicrobial Resistance and Use Surveillance System (GLASS) database on the country enrollments and National Action Plans (NAPs) were collected from the World Health Organization Regional Office for Africa [40,57].

GLASS has a broad scope, aiming to standardize data collection, analysis, interpretation, and sharing among countries. It supports capacity building and monitors national surveillance systems. GLASS encourages transitioning from laboratory-based surveillance to a comprehensive system that includes epidemiological, clinical, and population-level data. It intends to incorporate data from monitoring AMR in humans, the food chain, and the environment. GLASS organizes AMR surveillance activities into technical modules, using routinely available data and targeted activities to fulfill specific needs. Additionally, GLASS offers support through evidence-based guidelines and technical resources to help countries and regions enhance their capabilities and take necessary corrective actions.

Both data sources were interpolated to provide a broader picture of AMR and antimicrobial consumption and use (AMC/U) surveillance systems in West Africa.

## 5. Future Directions and Recommendations for AMR in West Africa

Based on this review, many recommendations can be derived for improving AMR surveillance in West Africa, including addressing the limited capacity to generate, collect, analyze, and report data. Efforts should be directed toward strengthening healthcare systems in the region to carry out these activities effectively. This can be achieved through WAHO targeted training programs, quadripartite (FAO, WOAH, WHO, UNEP) targeted training programs, resource allocation, and collaboration with international partners. Additionally, emphasis should be placed on improving the quality of AMR surveillance data by ensuring cleanliness, completeness, representativeness, and reproducibility [58,59]. Standardized laboratory programs with a list of relevant pathogens (including bacteria, but also viruses (HIV and hepatitis) and fungi) to be tested and minimal required equipment/reagents should be adopted in alignment with international standards and guidelines and implemented [60]. Standardized data collection tools should also be developed and implemented, personnel should be trained, and regular quality assurance assessments should be conducted [61].

Another recommendation is to enhance the capacity to use data for decision making. Investing in training healthcare professionals and policymakers on how to interpret and utilize AMR surveillance data for decision making is crucial [62,63,64,65]. This requires various capacity-building activities, including training programs, workshops, and the development of user-friendly data visualization and analysis tools. Moreover, strengthening laboratory diagnostic capacities is essential for the accurate and timely detection of antimicrobial resistance [62,66]. This involves improving access to quality laboratory facilities, training laboratory personnel, and ensuring the availability of necessary equipment and reagents [62,66]. Laboratories should be encouraged to implement a Quality Management System [60].

Monitoring and evaluation (M&E) plans for NAPs for AMR should be prioritized as well as strategies to address stockouts of essential antimicrobial medications and reagents [67,68]. NAPs should include robust M&E plans to assess progress and identify areas for improvement [67,68]. Strengthening the M&E mechanisms of NAPs is necessary for tracking progress and informing decision making. Countries must finalize and adopt their NAPs for AMR and, if needed, revise them to align with current global guidelines and priorities, ensuring well-represented multi-stakeholder committees [69]. Ensuring a consistent supply of essential antimicrobial medications and reagents is vital to combat resistance effectively. Countries have increasing limitations in procuring medicines, antimicrobials, and AST disks. Therefore, a pooled procurement mechanism for West African states could alleviate countries’ challenges in procuring these supplies individually [70,71].

Additionally, adopting and implementing standardized guidelines for AST is recommended to promote consistency, comparability, and accuracy across healthcare facilities and laboratories [72,73]. ECOWAS countries should also ensure the adoption and implementation of standardized policies and guidelines, promoting data use for decision making at all levels through the WAHO and the Regional Center for Surveillance and Diseases Control (RCSDC) mandate. For example, countries should consider revising their Essential Medicines List to include the WHO AwARe Categorization, updating the National Medicines Policy to include AMR where this is lacking as well as with national health policies and national antimicrobial stewardship guidelines to support interventions to optimize antimicrobial use at national and healthcare facility levels.

Furthermore, countries should enroll in GLASS if not already enrolled and aim to submit AMR surveillance data to GLASS in 2023 [40,66]. Multisectoral coordination among government entities, non-governmental organizations, and the private sector should be enhanced to address AMR comprehensively. Recognizing and leveraging existing strengths in AMR surveillance in West Africa is crucial, as countries should share best practices, foster collaboration, and facilitate knowledge exchange among themselves [69].

Given the region’s diverse healthcare systems and policies, a one-size-fits-all approach would be inappropriate. Yet, certain core principles about antimicrobial stewardship, drug quality, and patient care standards are relevant across clinical settings. Harmonizing regulations should be made through standardized AMR policies and a supportive legislative framework, accommodating the national health systems’ capacities and challenges. Strengthening surveillance systems by establishing a regional network for monitoring AMR patterns and integrating databases for data analysis will aid in identifying trends and outbreaks and facilitate cross-border collaboration with regional AMR committees. Engaging various stakeholders, including governments, healthcare providers, pharmaceutical companies, civil society organizations, and the private sector, is necessary to ensure sustainable coordination and a comprehensive approach. To effectively implement the aforementioned recommendations and establish a sustainable strategy for tackling AMR and its associated challenges, it is imperative to underscore the necessity of allocating local resources. It is crucial to prioritize local resource mobilization and secure dedicated local financing for AMR initiatives. Thus, the emphasis on local financial commitment is indispensable, as reliance solely on external support may not be sufficient or sustainable in the long run. This should also be coupled with the promotion of research in AMR in various areas, including prevalence and cost-effectiveness studies in West Africa and LMICs to enrich the knowledge database to inform evidence-based interventions.

## 6. Conclusions

In conclusion, the present review utilized data from the TrACSS platform to assess the general statistics of AMR surveillance in the ECOWAS countries in 2022. The study overviews the participating countries’ surveillance systems and activities. While there have been notable achievements in AMR surveillance and action planning in West Africa, significant gaps and challenges still need to be addressed. Strengthening surveillance systems, improving laboratory capacities, enhancing data integration and use, and promoting multisectoral collaboration are critical areas that require the highest attention. Continued support, technical assistance, and resource allocation are crucial to sustain and accelerate progress in tackling AMR in the region. Mobilizing local resources and establishing a regional pool procurement mechanism for essential supplies for combating AMR could be a cornerstone of the solution. These findings contribute to understanding the current state of AMR surveillance in the ECOWAS region. They can inform future interventions and policies addressing antimicrobial resistance in West Africa. By implementing the above-mentioned recommendations, West African countries can strengthen their AMR surveillance efforts, enhance their capacity to combat antimicrobial resistance, and contribute to global initiatives addressing this critical public health challenge.

## 7. Limitations

This review has several limitations. Firstly, the analysis was based on data from the TrACSS and GLASS databases, which might not represent all AMR surveillance activities in the ECOWAS countries. Additionally, the data were self-reported by participating countries, and variations in surveillance methodologies and reporting systems may exist. Another limitation is that TrACSS responses do not permit an assessment of the adequacy and impact of the activities covered. While a response of “No” is always significant, a response of “Yes” provides no granularity on the strengths or weaknesses of the activities. While several laboratories may follow certain guidelines, there is currently no technical assessment of how efficiently, strictly, and correctly they follow them. The WAHO is planning to conduct missions to evaluate and reinforce this. Finally, the analysis focused on general descriptive information on the surveillance system. It did not include an in-depth analysis of specific AMR trends or factors influencing AMR patterns in the ECOWAS region.

## Figures and Tables

**Figure 1 antibiotics-13-00627-f001:**
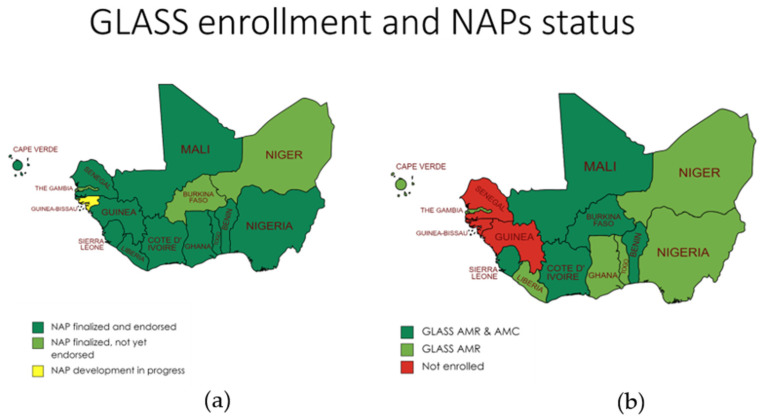
(**a**) is the NAP status, (**b**) is the GLASS enrollment status.

**Figure 2 antibiotics-13-00627-f002:**
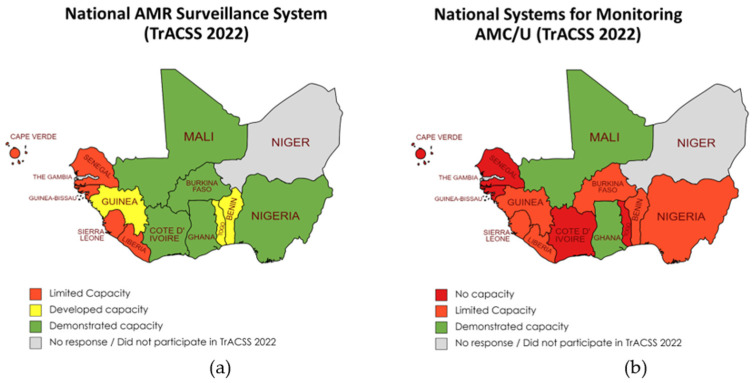
(**a**) is the status of the national AMR surveillance system, and (**b**) is the status of the national systems for monitoring AMC/U.

**Figure 3 antibiotics-13-00627-f003:**
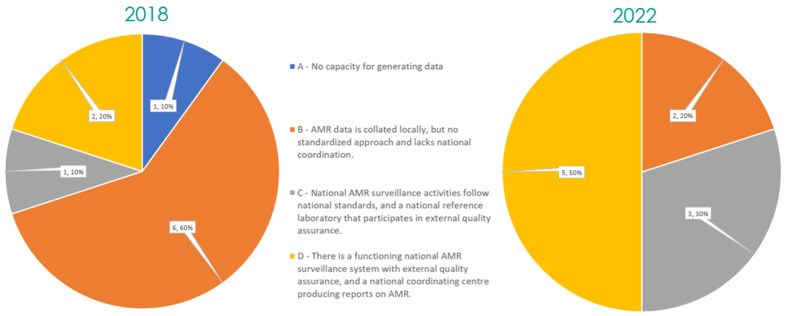
The AMR surveillance capacity in 2018 vs. 2022.

**Figure 4 antibiotics-13-00627-f004:**
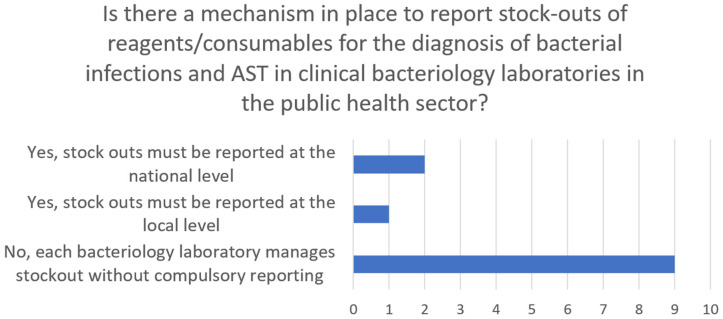
The reporting mechanism for stockout.

**Table 1 antibiotics-13-00627-t001:** Country response status and background information for West African countries.

	TrACSS 2018	TrACSS 2022	Income Group *
**Benin**	Yes	Yes	Lower-Middle Income
**Burkina Faso**	Yes	Yes	Low Income
**Cabo Verde**	No	Yes	Lower-Middle Income
**Côte d’Ivoire**	Yes	Yes	Lower-Middle Income
**Gambia**	No	No	Low Income
**Ghana**	Yes	Yes	Lower-Middle Income
**Guinea**	Yes	Yes	Lower-Middle Income
**Guinea-Bissau**	No	Yes	Low Income
**Liberia**	Yes	Yes	Low Income
**Mali**	Yes	Yes	Low Income
**Niger**	No	Yes †	Low Income
**Nigeria**	Yes	Yes	Lower-Middle Income
**Senegal**	No	Yes	Lower-Middle Income
**Sierra Leone**	Yes	Yes	Low Income
**Togo**	Yes	Yes	Low Income

* The World Bank classifies low-income economies as those with a Gross National Income (GNI) per capita less than $1135, and the lower middle-income economies are those with a GNI per capita between $1136 and $4465 [39]. † Incomplete response.

## Data Availability

The data sources have been cited in the methods.

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
