# Peer review of "Surveillance of Antimicrobial Resistance in the ECOWAS Region: Setting the Scene for Critical Interventions Needed"

_antibiotics, 2024, doi:10.3390/antibiotics13070627_

Round 1

Reviewer 1 Report

Comments and Suggestions for Authors

This review provides a comprehensive introduction to the development of antibiotic resistance and resistance monitoring systems in the ECOWAS region (ECOWAS countries). Especially in the "Results" section, the comprehensive display of relevant drug resistance monitoring information in the ECOWAS region provides important information for other regions to understand antibiotic resistance information in ECOWAS countries and engage in cooperation. This review also places drug resistance monitoring in West Africa under the One health framework, with a comprehensive focus on monitoring AMR in humans, the food chain, and the environment.

The parts of “recommendations for AMR in West Africa” are based on the review above, but those should not take up a lot of space as the content of this review. It is recommended to delete it and keep a paragraph related to the discussion section.

Author Response

Comment 1: This review provides a comprehensive introduction to the development of antibiotic resistance and resistance monitoring systems in the ECOWAS region (ECOWAS countries). Especially in the "Results" section, the comprehensive display of relevant drug resistance monitoring information in the ECOWAS region provides important information for other regions to understand antibiotic resistance information in ECOWAS countries and engage in cooperation. This review also places drug resistance monitoring in West Africa under the One health framework, with a comprehensive focus on monitoring AMR in humans, the food chain, and the environment.

Response: Thank you very much for your insightful comment.

Comment 2: The parts of “recommendations for AMR in West Africa” are based on the review above, but those should not take up a lot of space as the content of this review. It is recommended to delete it and keep a paragraph related to the discussion section.

Response: Thank you very much for your comments. We have updated the title to "Future directions and recommendations." this aligns with the recommended sections for a review in the journal's author guidelines https://www.mdpi.com/journal/antibiotics/instructions. Furthermore, we believe this is one of the most important sections in our article as it provides a way forward and puts forward actionable takeaway messages that would benefit the readers.

Reviewer 2 Report

Comments and Suggestions for Authors Dear authors, 
  • you should add section on the impact of COVID-19 on antimicrobial stewardship/antimicrobial resistance, there is gap here need to be addressed
  • obviously, this article for ECOWAS regions, however, differences in healthcare system and policies exist, you need to elaborate how would you unify efforts to tackle antimicrobial resistance/antimicrobial stewardship?
  • you need to add section in your article about the economic impact and discuss the cost-effectiveness of antimicrobial stewardship in ECOWAS region
  • After writing your comprehensive article, you need to compare and contrast comparable effects for antimicrobial resistance/antimicrobial stewardship done elsewhere in the world. for example, North America, Middle East, etc

Other minor edits suggested

1- I change "approach" to "method" and move it after introduction

2- indicate in the title study type "review."

3- The limitation should be the last paragraph of the discussion 

Comments on the Quality of English Language

None

Author Response

Thank you very much for your comments.

Comment 1: you should add section on the impact of COVID-19 on antimicrobial stewardship/antimicrobial resistance, there is gap here need to be addressed

Response: Thank you for your comment. We have added a paragraph on the effect of the COVID-19 pandemic on AMR.

"The COVID-19 pandemic has impacted AMR in multiple ways, including reversing some of the progress made in the fight against AMR worldwide. (Langford et al., 2023; CDC, 2024). Antibiotics, especially broad-spectrum ones, were overprescribed due to challenges distinguishing bacterial infections from COVID-19, which was exacerbated by the use of medical devices and shortages of personal protective equipment (Rayan, 2023). The pandemic also underscored the importance of improving AMR management, with increased awareness of hygiene practices and infection prevention and control measures and the need to strengthen laboratory capacities and surveillance systems (Poudyal et al., 2023; Rayan, 2023). The pandemic also highlighted the necessity to optimize antibiotic use, including appropriate prescribing, rapid diagnostics, and adherence to treatment guidelines (Rayan, 2023). Additionally, vaccinations can reduce the need for antibiotics, and continued investment in AMR research, surveillance, and stewardship programs is necessary to address this growing threat (Rayan, 2023)."

Comment 2: obviously, this article for ECOWAS regions, however, differences in healthcare system and policies exist, you need to elaborate how would you unify efforts to tackle antimicrobial resistance/antimicrobial stewardship?

Response: 

"Given the region's diverse healthcare systems and policies, a one-size-fits-all approach would be inappropriate. Yet, certain core principles about antimicrobial stewardship, drug quality, and patient care standards are relevant across clinical settings. Harmonizing regulations should be made through standardized AMR policies and a supportive legislative framework, accommodating the national health systems' capacities and challenges. Strengthening surveillance systems by establishing a regional network for monitoring AMR patterns and integrating databases for data analysis will aid in identifying trends and outbreaks and facilitate cross-border collaboration with regional AMR committees. Engaging various stakeholders, including governments, healthcare providers, pharmaceutical companies, civil society organizations, and the private sector, is necessary to ensure sustainable coordination and a comprehensive approach."

Comment 3: you need to add section in your article about the economic impact and discuss the cost-effectiveness of antimicrobial stewardship in ECOWAS region

Response: Thank you very much for your valuable comment. We have added the below in the recommendations and introdutions. 

"two cohort studies from Ghana have shown that AMR is associated with an additional mean healthcare providers’ annual cost of 650,000 USD and a patient cost of 1.4 million USD, calling for the prioritization of the AMR prevention, surveillance, and mitigation in hospitals and the community (Otieku, Fenny, et al., 2023; Otieku, Kurtzhals, et al., 2023). According to the World Bank’s modeled estimates, AMR health costs could increase to $330 billion under a low burden and $1.2 trillion under a low burden(Ahmed et al., 2017)."

"This should also be coupled with the promotion of research in AMR in various areas, including prevalence and cost-effectiveness studies in West Africa and LMICs to enrich the knowledge database to inform evidence-based interventions."

Comment 4: After writing your comprehensive article, you need to compare and contrast comparable effects for antimicrobial resistance/antimicrobial stewardship done elsewhere in the world. for example, North America, Middle East, etc

Response: Thank you very much for your great comment. However, this article reviews AMR surveillance experiences, policies, and surveillance systems in ECOWAS countries, and thus, a thorough comparison with the rest of the world is beyond the scope of this paper. We include several statements that address your comments and contextual information that shows that ECOWAS and Africa are different from the rest of the world. In the introduction, we compare West Africa to other regions. 

Furthermore, we have also enriched our discussions with many references from outside the region that share similar recommendations for AMR surveillance.

"According to a predictive model, bacterial AMR was associated with 4.95 million deaths in 2019, and the most significant burden occurred in the sub-Saharan Africa Region, where 1.07 million people died because of bacterial antimicrobial resistance (Murray et al., 2020). The burden of AMR is not evenly distributed, and resource-limited countries face a higher impact (Okeke et al., 2005; Laxminarayan et al., 2013; Bebell and Muiru, 2014). In Africa, the risks associated with AMR are amplified by the high prevalence of infectious diseases and fragile healthcare systems (Iwu-Jaja et al., 2021; Sartorius et al., 2024). Furthermore, this is exacerbated by the misuse and overuse of antibiotics, including non-prescription use, which is widespread in some parts of Africa and exceeds 90% of all antimicrobial use in some parts of Africa (Morgan et al., 2011). While considerable progress has been made in understanding the drivers of AMR and implementing antibiotic stewardship programs, low- and middle-income countries have often been left behind from such advancements."

"According to a systematic review of AMR by the WHO Regional Office for Africa in 2021, there is a significant lack of data on the prevalence of AMR in the WHO African Region, particularly in West Africa."

We added further references to support our discussion and recommendations with publications from different countries.

Minor comment 1: I change "approach" to "method" and move it after introduction.

Response: Thank you for pointing this out. We have updated the heading accordingly.

Minor comment 2:  indicate in the title study type "review."

Response: We do have the study type set as review and mentioned directly before the title. We want to avoid repetition, which would make the title more lengthy.

Minor comment 3: The limitation should be the last paragraph of the discussion 

Response: Thank you for pointing this out. We have updated the heading accordingly.

Reviewer 3 Report

Comments and Suggestions for Authors

In the present manuscript, the authors provide an overview of antimicrobial resistance surveillance in West African countries. While the manuscript addresses an extremely important health topic, which is especially crucial for countries with less developed healthcare systems, it has several limitations acknowledged by the authors. These limitations impact the overall quality of the manuscript. Nevertheless, I believe it offers a valuable perspective on the current situation in West Africa.

The title of chapter 2.2 is likely incorrect.

Author Response

Comment 1: In the present manuscript, the authors provide an overview of antimicrobial resistance surveillance in West African countries. While the manuscript addresses an extremely important health topic, which is especially crucial for countries with less developed healthcare systems, it has several limitations acknowledged by the authors. These limitations impact the overall quality of the manuscript. Nevertheless, I believe it offers a valuable perspective on the current situation in West Africa.

Response 1: Thank you very much for your insightful comment. 

Comment 2:The title of chapter 2.2 is likely incorrect.

Response 2: Thank you for pointing this out. We have updated the title to "GLASS enrollment Status"

Round 2

Reviewer 1 Report

Comments and Suggestions for Authors

The manuscript has been sufficiently improved to warrant publication in Antibiotics.